# Effectiveness of interactive teaching intervention on medical students' knowledge and attitudes toward stem cells, their therapeutic uses, and potential research applications

Fayez Abdulrazeq[1], Khalid A. Kheirallah[1], Abdel-Hameed Al-Mistarehi[1], Samir Al Bashir[2], Mohammad A. ALQudah[2], Abdallah Alzoubi[3,4], Jomana Alsulaiman[5], Mazhar S. Al Zoubi[6] and Abdulwahab Al-Maamari[7]

[1] Department of Public Health and Family Medicine, Faculty of Medicine, Jordan University of Science and Technology, Irbid, Jordan

[2] Department of Pathology and Microbiology, Faculty of Medicine, Jordan University of Science and Technology, Irbid, Jordan

[3] College of Medicine, Ajman University of Science & Technology, Ajman, United Arab Emirates

[4] Department of Pharmacology, Faculty of Medicine, Jordan University of Science and Technology, Irbid, Jordan

[5] Department of Pediatrics, Faculty of Medicine, Yarmouk University, Irbid, Jordan

[6] Department of Basic Medical Sciences, Faculty of Medicine, Yarmouk University, Irbid, Jordan

[7] Department of Private Law, Faculty of Law, Al-Isra Private University, Amman, Jordan

Corresponding authors
Khalid A. Kheirallah,
kakheirallah@just.edu.jo
Abdel-Hameed Al-Mistarehi,
awalmistarehi18@med.just.edu.jo

## ABSTRACT

**Background**. Stem cell science is rapidly developing with the potential to alleviate many non-treatable diseases. Medical students, as future physicians, should be equipped with the proper knowledge and attitude regarding this hopeful field. Interactive teaching, whereby the teachers actively involve the students in the learning process, is a promising approach to improve their interest, knowledge, and team spirit. This study aims to evaluate the effectiveness of an interactive teaching intervention on medical students' knowledge and attitudes about stem cell research and therapy.

**Methods**. A pre-post test study design was employed. A six-session interactive teaching course was conducted for a duration of six weeks as an intervention. Pre- and post-intervention surveys were used. The differences in the mean scores of students' knowledge and attitudes were examined using paired t-test, while gender differences were examined using an independent t-test.

**Results**. Out of 71 sixth-year medical students from different nationalities invited to participate in this study, the interactive teaching course was initiated by 58 students resulting in a participation rate of 81.7%. Out of 58 students, 48 (82.8%) completed the entire course. The mean age (standard deviation) of students was 24 (1.2) years, and 32 (66.7%) were males. The results showed poor knowledge about stem cells among the medical students in the pre-intervention phase. Total scores of stem cell-related knowledge and attitudes significantly improved post-intervention. Gender differences in knowledge and attitudes scores were not statistically significant post-intervention.

**Conclusions**. Integrating stem cell science into medical curricula coupled with interactive learning approaches effectively increased students' knowledge about recent
advances in stem cell research and therapy and improved attitudes toward stem cell research and applications.

## INTRODUCTION

The emerging stem cell (SC) biology discipline and the rapid revolution in SC research have radically transformed our thinking of cells, evolution, and disease. Using SCs for clinical applications represents the future of translational medicine since SCs can potentially be used to treat many kinds of difficult diseases that cannot currently be treated (*Chang et al., 2018*; *Protze, Lee & Keller, 2019*). Advances in SC research combined with tissue engineering techniques promise therapies to restore or replace damaged tissues (*Kwon et al., 2018*). This raises the need for medical education to introduce basic SC knowledge and the concept of translational medicine into the life sciences field. At the same time, SC research and applications still raise complex social, legal, ethical, and religious issues (*Al-Aqeel, 2005*; *Curley & Sharples, 2006*; *Pourebrahim, Goldouzian & Ramezani, 2020*), especially in conservative societies (*Bouzenita, 2017*).

The emerging developments in SC applications are transforming the priorities of undergraduate and postgraduate medical educational programs (*Scott, 2015*). Today, the traditional academic model for medical education is challenged by an evident gap between the rapidly changing disciplines in basic biomedical sciences and clinical practice. Although medical students have access to SC research theoretical advancements, traditional teaching approaches still fail to bridge this practice gap (*Brass, 2009*). Thus, updated teaching techniques that facilitate the integration of SC research advancements with clinical practice are critical for medical students to achieve optimum patient care (*Knoepfler, 2013*). Restructuring medical education to meet the current and future health care needs of SC-based interventions, including new curricula featuring the ethical, legal, and social implications of SC research, are thus a priority (*Pershing & Fuchs, 2013*; *Pierret & Friedrichsen, 2009*).

Since the early 1990s, many medical curricula have transitioned from traditional subject-based teaching toward integrated system-based teaching (*Ling et al., 2008*). Traditional didactic lectures for one hour become monotonous after 15–20 min as students' participation in the learning process is minimal (*Gupta, BhattiK & AgnihotriP, 2015*). On the other hand, the interactive teaching approach actively engages learners and interchanges ideas between learners and facilitators (*Kaur et al., 2011*). The effectiveness of educational interventions in increasing knowledge and attitudes towards SC applications was reported previously by a few studies (*Azzazy & Mohamed, 2016*; *Jin et al., 2018*; *Kaya et al., 2015*).

Although it is currently a hot research topic, SC education for students is uncommon (*Pierret & Friedrichsen, 2009*). The interactive teaching modality was designed to introduce

medical students to the pioneering area of SC biology and shed light on current advances in SC research. Medical students, as future physicians, are expected to answer patients' questions regarding SCs and help them differentiate between what is realistic and unrealistic regarding SC-based therapies. Also, they should be able to use evolving discoveries in SC research and apply them in the care of patients. Thus, we aimed in this study to gauge the medical students' knowledge and attitudes toward SCs, their therapeutic uses, and potential research applications and then evaluate the effectiveness of a six-session interactive teaching intervention on their knowledge and attitudes.

## MATERIAL AND METHODS

### Study design, participants, and setting

A pre-post test design was employed for a sample of 71 sixth-year medical students, at the University of Science and Technology Yemen-Jordan branch (USTY-Jo), during the first semester of the academic year 2018-2019. An orientation lecture was held before the initiation of the study to explain the study aims, design, and details for the students and invite the students to participate. Study participation was voluntary, and the pre-intervention survey was distributed to all medical students who agreed to participate. After that, the participants were then invited to attend a six-session interactive teaching course, the intervention, for a duration of six weeks. This intervention was a part of phase I of the "Stem Cells: Hope or Hype?" project. Each interactive session lasted two to three hours and included brainstorming, learning by teaching, role-playing, class debate, panel discussions, reflections on stories, real-life situations, case-based scenarios, or videos. Details about the intervention are summarized in Table 1. After finishing the intervention, the same survey was distributed among the participants.

### Ethical considerations

The study protocol was reviewed and ethically approved by the Institutional Review Board (IRB) of the research and ethics committee at USTY-Jo (IRB number, 9/120/2019). This study was conducted following the 1975 Helsinki declaration, as revised in 2008 and later amendments or comparable ethical standards. The study objectives and design were duly explained to the study participants during the orientation lecture and with each intervention session. They were informed about the study objectives, design, duration, interactive teaching methods, and the guest lecturers were invited to participate in the interactive teaching sessions. A written, signed, informed consent was obtained from each participant. Participants were informed that they could terminate the survey and interventions at any time desired. Participants did not receive any compensation or rewards for their participation in the study.

To conduct the study with keeping the participant anonymity and survey confidentiality in light of its pre-post test design, we need a method of identifying the participant so that we can measure the change from the first survey to the second for the same participant without breaching the anonymity and confidentiality. One possible way is to anonymously generate a unique ID code for each participant. The codes should be easily recovered if needed, unlikely duplicated across multiple respondents, and unique for each participant. Thus,

**Table 1 Detailed study intervention.**

| Number and title of intervention week | Objectives of the intervention | Interactive teaching methods |
|---|---|---|
| **Week one:** Stem cell basic biology | ✓ Reviewing the history of stem cell (SC) research. ✓ Understanding the basic biology of SCs and identifying characteristics that distinguish SCs from other types of cells. ✓ Classifying SCs according to source and potency. | **Brainstorming:** The lecturer asked students an opening question: what do you know about SCs?, then he used the whiteboard to list all the ideas generated by the students and grouped them into few headlines. **Visual aids:** The lecturer presented a short video about the discovery of the microscope by Robert Hooke, and then he presented a diagram illustrating major historical events in SC research. |
| **Week two:** Stem cell potential applications | Recognizing potential applications of SCs in studying early human development, modeling diseases in a culture dish, testing new drugs, and restoring lost tissues. | **Group activity and learning by teaching:** Students were divided into eight groups and were given one of four topics that cover potential applications of SCs. Each group had to read five articles about the topic and do a seminar for other students. |
| **Week three:** Unproven stem cell therapies and stem cell tourism | ✓ Listing current therapeutic uses of SCs such as bone marrow transplantation for leukemia. ✓ Shedding light on potential therapeutic uses of SCs such as limbal SCs for degenerative eye diseases. ✓ Increasing awareness about SC tourism and severe risks due to trying unproven SC therapies. | **Case-based scenarios:** For patients who tried unproven SC therapies. **Group activity:** Students were divided into eight groups assigned to search for websites that promote unproven SC therapies. |
| **Week four:** Stem cell research | ✓ Understanding the induced pluripotent stem cells (iPSCs) and the role of transcription factors. ✓ Explaining SC-assisted technologies such as MRT, SCNT, and human/animal chimeras. | **Story:** A reflection on Shinya Yamanaka's story, who won the Nobel Prize for discovering induced pluripotent SCs. |
| **Week five:** Cord blood banking and donation | ✓Explaining techniques and procedures of cord blood collection, banking, and donation. ✓ Summarizing advantages and disadvantages of cord blood transplantation in comparison with bone marrow transplantation. ✓ Comparing between different types of cord blood banks. | **Role-playing:** Students played different roles assigned to them: parents who are interested in cord blood banking and healthcare providers who should answer parents' questions. **Guest lecturer:** To take about cord blood banking. **Real-life situations:** Students provided health education for pregnant women about cord blood banking. |
| **Week six:** Bioethics of stem cell research | Discussing ethical controversies surrounding SC research and their-assisted technologies. | **Panel discussion:** With bioethics expert. **Class debate:** Class was divided into eight groups; four groups argued for another four groups against research involving embryonic SCs. |

a coding system was created based on the participants' names and birth dates. We asked each participant to generate their own ID code by providing the first letters of their first and family names (A–Z) plus a four-digit code that represents birthday (01-31) and month of birth (01-12). The participants' ID codes were essential for data analysis to compare pre-and-post intervention scores and avoid duplicated data with preserving anonymity. Thus, the study was undertaken with complete confidentiality, and information provided by study participants was not disclosed to others.

## Study tools

After detailed reviewing the literature regarding SC knowledge and attitudes, the researchers developed a structured, self-administered questionnaire. The questionnaire was not based on a particular study but preferably on information from various studies and recent guidelines from international organizations such as the International Society for Stem Cell Research (ISSCR) and the New York Stem Cell Foundation (NYSCF) (*Azzazy & Mohamed, 2016*; *Lovell-Badge et al., 2021*; *NYSCF, 2017*). The questionnaire was reviewed by a panel of experts in SC clinical practice and teaching, pilot-tested on 20 participants, and the necessary modifications were done. The questionnaire was designed and distributed in the English language as it is the official teaching language of the Jordanian medical schools. A soft copy of the distributed questionnaire is provided in File S1.

The questionnaire was started by asking the participant to provide their ID code following specific instructions, as mentioned earlier. Then, the questionnaire included three major sections: demographic characteristics, SC knowledge, and SC attitudes. The demographics section included questions about age, gender, nationality of participants, name of the registered medical school, and student year level. The SC knowledge section began with a rating question about the participant perception of knowledge regarding SCs in general with a ten-point Likert scale, ranging from "zero = low knowledge" to "10 = high knowledge". Then, a question about the participants' preferred sources of knowledge about SCs with multiple choices included books, medical journals, workshops, social media, lectures, medical conferences, panel discussions, and other sources.

After that, the SC knowledge section included 27 statements to measure the participants' knowledge regarding SCs. These 27 statements consist of eleven correctly stated statements, seven false or misleading statements, and neither true nor false statements, with a total of nine statements. The SC knowledge section statements have an acceptable to excellent internal consistency and reliability with a Cronbach's Alpha of 0.61 and 0.78 in pre-and post-intervention, respectively. The SC knowledge section was divided into four domains, including basic knowledge about SCs with a total of 13 statements (Cronbach's Alpha =0.42 and 0.61 in pre-and post-intervention, respectively), potential applications of SCs with a total of four statements (Cronbach's Alpha =0.69 and 0.66), therapeutic uses of SCs with a total of four statements (Cronbach's Alpha =0.44 and 0.32), and lastly the participant knowledge about SC research with a total of six statements (Cronbach's Alpha =0.86 and 0.75). The third section was designed to assess the medical students' attitudes toward SCs *via* a total of ten statements with Cronbach's Alpha of 0.76 and 0.68 in pre-and post-intervention, respectively.

Participants responded to each statement of the SC knowledge and attitude scales described above, using a 5-point Likert scale ranging between "Strongly Disagree" and "Strongly Agree" for each statement to provide high-resolution data and detailed information. After that, each response was scored from "Zero = Strongly Disagree" to "Four = Strongly Agree" except for the seven false statements, where the code was reversed to be "Four = Strongly Disagree" and "Zero = Strongly Agree". Responses to statements were summed to create scores for the total knowledge, each of the four knowledge domains, and total attitude. Thus, higher scores indicated good knowledge and positive attitude, while lower scores indicated poor knowledge and negative attitude. Knowledge scores ranged from zero to 108 for "total SC knowledge", zero to 52 for "SC basic knowledge", zero to 16 for "SC potential applications", zero to 16 for "SC therapeutic uses", and zero to 24 for "SC research". The total attitude score ranged from zero to 40. After that, the scores of scales were converted into mean scores ranging from zero to four by dividing the scale score on the number of scale statements.

## Statistical analysis

Data were analyzed using IBM Statistical Package for Social Sciences (SPSS), Windows Version 25.0 (IBM Corp., Armonk, NY, USA). Internal consistency for scales and subscales were tested using Cronbach's alpha. Descriptive statistics were presented as means and standard deviations (SD) for continuous variables after verifying the normality of the dataset. Categorical variables were presented as proportions and frequencies. A Paired-samples $t$-test was used to examine the mean differences (MD) in students' knowledge and attitude scores pre- and post-educational intervention, and the statistical significance and 95% confidence intervals of the difference in means were reported. Independent-samples $t$-test was used to examine mean gender differences in students' knowledge and attitude scores. A *p-value* was set at or less than *0.05* to be significant.

## RESULTS

Out of 71 sixth-year medical students invited to participate in this study, 58 initiated the interactive teaching course resulting in a participation rate of 81.7%. The final sample consisted of 48 medical students who initially enrolled and completed the entire six-week course sessions with a completion rate of 82.8%. Out of 48 medical students, 32 (66.7%) were males, and more than half (56.3%) were of Jordanian or Yemeni nationalities. The enrolled Students' mean age (SD) was 24.0 (1.2) years. Demographic characteristics of study participants are summarized in Table 2.

## Knowledge regarding stem cells

The three most common sources of knowledge regarding SCs before the intervention course were lectures (56.3%), social media (45.8%), and books (41.7%), while any participant did not report panel discussions as a source of knowledge (Fig. 1). Detailed information about pre-and post-educational intervention knowledge scores is summarized in Table 3. Pre-intervention, the lowest mean score among knowledge domains was observed with SC research section (1.76 (0.89)), and therapeutic uses (1.84 (0.63)), followed by SC basic
**Table 2 Respondents' Characteristics (n = 48).**

| Characteristics | Value |
|---|---|
| **Gender, *N (%)*** | |
| Male | 32 (66.7%) |
| Female | 16 (33.3%) |
| **Age, *M (SD)*** | 24.0 (1.2) |
| **Nationality, *N (%)*** | |
| Jordanian | 14 (29.2%) |
| Palestinian | 3 (6.3%) |
| Syrian | 8 (16.7%) |
| Iraqi | 6 (12.5%) |
| Yemeni | 13 (27.1%) |
| Others | 4 (8.3%) |

Notes.
Abbreviations: M, Mean; SD, Standard Deviation.

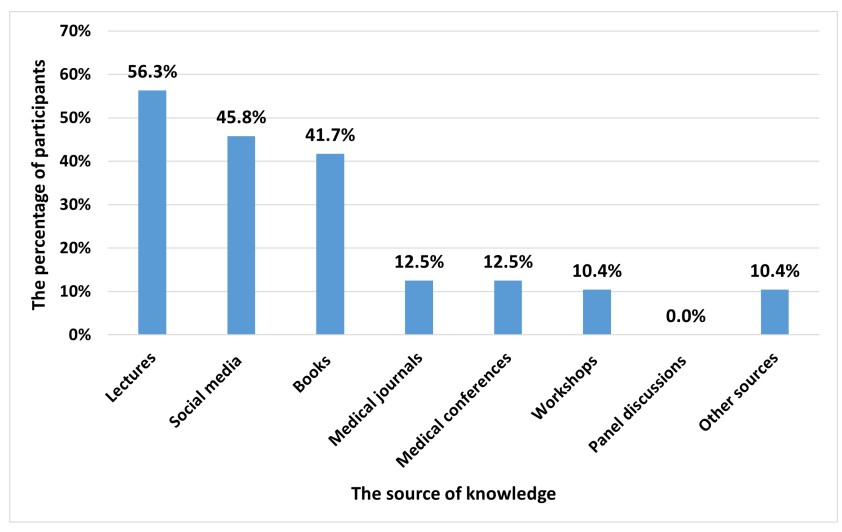

**Figure 1 The reported sources of knowledge about stem cells before the intervention course.**

knowledge (2.14 (0.30)) and their potential applications (2.66 (0.77)). The mean scores of all knowledge domains were lower than three in the pre-intervention phase of the study. The mean (SD) total knowledge score was 2.09 (0.30) pre-intervention, which is significantly improved to 3.09 (0.41) post-intervention ($p < 0.001$). Similarly, all knowledge domains' scores significantly increased following the intervention.

The mean SC basic knowledge domain score significantly increased from 2.14 (0.30) pre-intervention to 3.09 (0.47) post-intervention ($p < 0.001$). Post-intervention *vs.* pre-intervention, participants reported improved the knowledge with different types of SCs (3.77 (0.43) *vs.* 1.88 (0.94), $p < 0.001$), sources of SCs (3.67 (0.52) *vs.* 2.17 (0.78), $p < 0.001$), therapeutic uses of SCs (3.69 (0.51) *vs.* 2.15 (1.05), $p < 0.001$) and three germ layers from which tissues and organs are generated (3.69 (0.59) *vs.* 2.67 (1.02), $p < 0.001$). Students'

**Table 3  Pre- and post-educational intervention mean knowledge scores and differences ($n = 48$).**

| Pre- and post-educational intervention mean scores of students' knowledge regarding stem cells, their potential applications, therapeutic uses, and research involving them | | | | Differences between pre-and post-educational interventions | | | |
|---|---|---|---|---|---|---|---|
| | Score | M (SD) | Min–Max | MD | 95% CI | | p-value |
| | | | | | Lower | Upper | |
| Stem cells: basic knowledge | | | | | | | |
| 1- I have sufficient knowledge of different types of stem cells, such as adult and embryonic stem cells. | Pre | 1.88 (0.94) | 0–4 | 1.89 | 1.57 | 2.23 | <0.001* |
| | Post | 3.77 (0.43) | 3–4 | | | | |
| 2- I have sufficient knowledge of the sources of stem cells. | Pre | 2.17 (0.78) | 0–4 | 1.50 | 1.22 | 1.78 | <0.001* |
| | Post | 3.67 (0.52) | 2–4 | | | | |
| 3- I have sufficient knowledge of the therapeutic uses of stem cells. | Pre | 2.15 (1.05) | 0–4 | 1.54 | 1.22 | 1.86 | <0.001* |
| | Post | 3.69 (0.51) | 2–4 | | | | |
| 4- I have sufficient knowledge of the three germ layers (endoderm, mesoderm and ectoderm), and organs and tissues generated from each layer. | Pre | 2.67 (1.02) | 0–4 | 1.02 | 0.70 | 1.34 | <0.001* |
| | Post | 3.69 (0.59) | 2–4 | | | | |
| 5- Cell differentiation is the process by which stem cells become more specialized cell types (true). | Pre | 2.85 (0.97) | 1–4 | 0.65 | 0.38 | 0.92 | <0.001* |
| | Post | 3.50 (0.72) | 1–4 | | | | |
| 6- As a stem cell differentiates, it gradually loses potency and becomes unipotent (true). | Pre | 2.23 (1.06) | 0–4 | 0.48 | 0.13 | 0.83 | 0.009* |
| | Post | 2.71 (1.32) | 0–4 | | | | |
| 7- Self-renewing is the ability of a stem cell to produce more stem cells with identical characteristics as the "parent" cell (true). | Pre | 2.46 (0.82) | 0–4 | 0.89 | 0.60 | 1.20 | <0.001* |
| | Post | 3.35 (0.84) | 1–4 | | | | |
| 8- Adult stem cells are pluripotent cells that have the potential to make all cell types of the body (false). † | Pre | 1.58 (1.16) | 0–4 | 1.34 | 0.75 | 1.92 | <0.001* |
| | Post | 2.92 (1.49) | 0–4 | | | | |
| 9- Bone marrow is the only source for adult stem cells (false).† | Pre | 2.17 (1.24) | 0–4 | 1.10 | 0.67 | 1.54 | <0.001* |
| | Post | 3.27 (1.25) | 0–4 | | | | |
| 10- Stem cells can differentiate into many cell types within a germ layer (true). | Pre | 2.73 (0.94) | 0–4 | 0.52 | 0.12 | 0.93 | 0.013* |
| | Post | 3.25 (1.02) | 0–4 | | | | |
| 11- Embryonic stem cells are derived from leftover blastocysts after in vitro fertilization (true). | Pre | 2.06 (0.76) | 0–4 | 0.90 | 0.48 | 1.31 | <0.001* |
| | Post | 2.96 (1.34) | 0–4 | | | | |
| 12- Embryonic stem cells are derived from the umbilical cord after childbirth (false). † | Pre | 1.35 (0.91) | 0–3 | 0.38 | −0.16 | 0.91 | 0.165 |
| | Post | 1.73 (1.65) | 0–4 | | | | |

**Table 3** (*continued*)

| Pre- and post-educational intervention mean scores of students' knowledge regarding stem cells, their potential applications, therapeutic uses, and research involving them | | | | Differences between pre-and post-educational interventions | | | |
|---|---|---|---|---|---|---|---|
| | Score | M (SD) | Min–Max | MD | 95% CI | | p-value |
| | | | | | Lower | Upper | |
| 13- Embryonic stem cells are derived from the trophoblast of blastocysts (false).[†] | Pre | 1.54 (0.65) | 0–4 | 0.25 | −0.25 | 0.75 | 0.316 |
| | Post | 1.79 (1.64) | 0–4 | | | | |
| **The total score of stem cell basic knowledge** | **Pre** | **2.14 (0.30)** | | **0.95** | **0.83** | **1.09** | **<0.001**[*] |
| | **Post** | **3.09 (0.47)** | | | | | |
| Stem cells: potential applications | | | | | | | |
| 14- Stem cells can be used to study early human development (true). | Pre | 3.02 (0.84) | 1–4 | 0.40 | 0.13 | 0.66 | 0.004[*] |
| | Post | 3.42 (0.71) | 1–4 | | | | |
| 15- Stem cells can be used to understand the pathophysiology and analyze disease mechanisms by modeling disease in a culture dish outside the human body (true). | Pre | 2.56 (1.09) | 0–4 | 0.92 | 0.48 | 1.35 | <0.001[*] |
| | Post | 3.48 (0.83) | 0–4 | | | | |
| 16- Stem cells can be used to test and screen new drug candidates and toxins to figure out their potential side effects (true). | Pre | 2.21 (1.09) | 0–4 | 1.27 | 0.84 | 1.70 | <0.001[*] |
| | Post | 3.48 (0.83) | 0–4 | | | | |
| 17- Stem cells can be used to replace or restore tissues that have been damaged by disease or injury, such as diabetes, heart attacks, Parkinson's disease, skin burns, or spinal cord injury (true). | Pre | 2.85 (1.09) | 0–4 | 0.73 | 0.38 | 1.08 | <0.001[*] |
| | Post | 3.58 (0.85) | 0–4 | | | | |
| **The total score of stem cell potential applications** | **Pre** | **2.66 (0.77)** | | **0.80** | **0.53** | **1.09** | **<0.001**[*] |
| | **Post** | **3.46 (0.59)** | | | | | |
| Stem cells: therapeutic uses | | | | | | | |
| 18- There is a wide range of conditions or diseases for which stem cell therapies have been proven to be safe and effective such as osteoarthritis and multiple sclerosis (false).[†] | Pre | 1.54 (0.97) | 0–4 | 0.69 | 0.23 | 1.15 | 0.004[*] |
| | Post | 2.23 (1.53) | 0–4 | | | | |
| 19- There is nothing to lose from trying unproven stem cell therapies since they can provide hope for hopeful patients (false). [†] | Pre | 1.71 (1.07) | 0–4 | 0.67 | 0.24 | 1.10 | 0.003[*] |
| | Post | 2.38 (1.39) | 0–4 | | | | |
| 20- Bone marrow-derived stem cells will spontaneously regenerate into different cell types such as hepatocytes and neural cells without manipulation in the lab (false). [†] | Pre | 1.88 (1.00) | 0–4 | 0.45 | −0.01 | 0.93 | 0.055 |
| | Post | 2.33 (1.53) | 0–4 | | | | |

**Table 3** (*continued*)

| Pre- and post-educational intervention mean scores of students' knowledge regarding stem cells, their potential applications, therapeutic uses, and research involving them | | | | Differences between pre-and post-educational interventions | | | |
|---|---|---|---|---|---|---|---|
| | *Score* | *M (SD)* | *Min–Max* | *MD* | 95% CI | | *p-value* |
| | | | | | Lower | Upper | |
| 21- If the balance is skewed between differentiation and self-renewing properties of stem cells, it may result in tumor formation (true). | Pre | 2.25 (0.79) | 0–4 | 0.63 | 0.26 | 0.99 | *<0.001*[*] |
| | Post | 2.88 (1.04) | 1–4 | | | | |
| The total score of stem cell therapeutic uses | **Pre** | **1.84 (0.63)** | | **0.61** | **0.36** | **0.85** | ***<0.001***[*] |
| | **Post** | **2.45 (0.80)** | | | | | |
| 22- I would be confident to explain the induced-Pluripotent Stem Cells (iPSCs). | Pre | 1.65 (1.16) | 0–4 | 1.75 | 1.38 | 2.12 | *<0.001*[*] |
| | Post | 3.40 (0.71) | 2–4 | | | | |
| 23- I would be confident to explain the transcription factors. | Pre | 1.85 (1.19) | 0–4 | 1.28 | 0.87 | 1.67 | *<0.001*[*] |
| | Post | 3.13 (0.89) | 0–4 | | | | |
| 24- Adult cells can be "reprogrammed" genetically to assume stem cell-like state (true). | Pre | 1.85 (1.05) | 0–4 | 1.46 | 1.04 | 1.88 | *<0.001*[*] |
| | Post | 3.31 (0.83) | 1–4 | | | | |
| 25- I would be confident to discuss the Somatic Cell Nuclear Transfer (SCNT). | Pre | 1.58 (1.07) | 0–4 | 1.48 | 1.07 | 1.89 | *<0.001*[*] |
| | Post | 3.06 (0.10) | 0–4 | | | | |
| 26- I would be confident to explain the differences between therapeutic cloning and reproductive cloning. | Pre | 1.81 (1.20) | 0–4 | 1.40 | 0.96 | 1.83 | *<0.001*[*] |
| | Post | 3.21 (0.82) | 0–4 | | | | |
| 27- I would be confident to discuss the mitochondrial replacement therapy. | Pre | 1.83 (1.19) | 0–4 | 1.69 | 1.28 | 2.10 | *<0.001*[*] |
| | Post | 3.52 (0.74) | 1–4 | | | | |
| **The total score of stem cell research** | **Pre** | **1.76 (0.89)** | | **1.51** | **1.20** | **1.82** | ***<0.001***[*] |
| | **Post** | **3.27 (0.56)** | | | | | |
| **The total knowledge score** | **Pre** | **2.09 (0.30)** | | **1.00** | **0.86** | **1.15** | ***<0.001***[*] |
| | **Post** | **3.09 (0.41)** | | | | | |

**Notes.**

*M*, Mean; *SD*, Standard Deviation; *Min*, Minimum score; *Max*, Maximum score; *CI*, Confidence Interval; *MD*, Mean Difference; *Pre*, Pre-educational intervention; *Post*, Post-educational intervention

[*] Significant at $p < 0.05$ based on paired-samples $t$-test.

[†] The code was reversed for the false or misleading statements to be "Four = Strongly Disagree" and "Zero = Strongly Agree".

The total score of knowledge is the sum of the statements' scores for the four major domains (stem cell basic knowledge, potential applications, therapeutic uses, and research) for each participant divided by 27.

knowledge of sources of embryonic SCs significantly improved for statements related to leftover blastocysts after *in vitro* fertilization (2.96 (1.34) post-intervention *vs.* 2.06 (0.76) pre-intervention, $p < 0.001$), but not for statements related to umbilical cord (1.73 (1.65) *vs.* 1.35 (0.91), $p = 0.165$) or trophoblast of blastocyst (1.79 (1.64) *vs.* 1.54 (0.65), $p = 0.316$).

For potential applications of SC domain, the mean score significantly increased from 2.66 (0.77) pre-intervention to 3.46 (0.59) post-intervention ($p < 0.001$). Post-intervention *vs.* pre-intervention, students reported significantly higher knowledge scores regarding potential applications of SCs such as replacing or restoring damaged tissues (3.58 (0.85) *vs.* 2.85 (1.09), $p < 0.001$), screening new drugs and toxins (3.48 (0.83) *vs.* 2.21 (1.09), $p < 0.001$), modeling disease in a culture dish (3.48 (0.83) *vs.* 2.56 (1.09), $p < 0.001$) and studying early human development (3.42 (0.71) *vs.* 3.02 (0.84), $p = 0.004$).

For SC therapeutic uses domain, the mean total score significantly increased from 1.84 (0.63) pre-intervention to 2.45 (0.80) after the intervention course ($p < 0.001$). Compared to pre-intervention results, the students became significantly more aware about the side effects of trying unproven SC therapies after the intervention course (2.38 (1.39) *vs.* 1.71 (1.07), $p = 0.003$) and tumor formation potential if the balance is skewed between cell differentiation and self-renewing properties of SCs (2.88 (1.04) *vs.* 2.25 (0.79), $p = 0.001$).

In the SC research domain, the mean total score significantly increased from 1.76 (0.89) to 3.27 (0.56) ($p < 0.001$). Post-intervention *vs.* pre-intervention, students became more comfortable in giving an explanation of induced pluripotent SCs (3.40 (0.71) *vs.* 1.65 (1.16), $p < 0.001$), transcription factors (3.13 (0.89) *vs.* 1.85 (1.19), $p < 0.001$), and differences between therapeutic cloning and reproductive cloning (3.21 (0.82) *vs.* 1.81 (1.20), $p < 0.001$). Moreover, participants became more knowledgeable that adult cells can be "reprogrammed" genetically to assume an SC-like state (3.31 (0.83) *vs.* 1.85 (1.05), $p < 0.001$). Students were also more comfortable discussing mitochondrial replacement therapy (3.52 (0.74) *vs.* 1.83 (1.19), $p < 0.001$) and somatic cell nuclear transfer (3.06 (0.10) *vs.* 1.58 (1.07), $p < 0.001$).

### Attitudes toward stem cells

As described in Table 4, the total attitude score significantly increased from 2.66 (0.56) to 2.85 (0.53) ($p = 0.048$). Post-intervention *vs.* pre-intervention, students became more interested in expanding their knowledge regarding SCs (3.77 (0.43) *vs.* 3.29 (0.92), $p = 0.001$), and considered a well-structured program or training focusing on SC science (3.48 (0.68) *vs.* 2.83 (0.91), $p < 0.001$). Students reported improved positive attitudes regarding integration of SC education in medical college curricula (3.35 (0.93) *vs.* 2.83 (0.10), $p = 0.010$), translational research (3.27 (0.84) *vs.* 2.83 (0.93), $p = 0.009$), and spending more money by government to support SC research (3.69 (0.72) *vs.* 3.38 (0.82), $p = 0.046$). In addition, participants' improvements in attitude were statistically significant towards umbilical cord blood donation (3.27 (1.13) *vs.* 2.85 (0.10), $p = 0.049$), but not for bone marrow donation (3.10 (1.23) *vs.* 2.81 (0.94), $p = 0.212$). Participants' negative attitudes regarding religious controversies surrounding SCs did not improve as the pre-intervention mean significantly decreased from 1.88 (1.10) to 1.13 (1.30), ($p = 0.003$). However, similar reductions reported in attitude mean scores related to ethical controversies surrounding SCs

**Table 4  Pre- and post-educational intervention mean attitude scores and differences ($n = 48$).**

| Statements | Score | M (SD) | Min–Max | MD | 95% CI | | p-value |
|---|---|---|---|---|---|---|---|
| | | | | | Lower | Upper | |
| 1- I am interested in expanding my knowledge about stem cells (positive). | Pre | 3.29 (0.92) | 0–4 | 0.48 | 0.22 | 0.74 | 0.001* |
| | Post | 3.77 (0.43) | 3–4 | | | | |
| 2- Stem cell education should be integrated into medical college curricula (positive). | Pre | 2.83 (0.10) | 0–4 | 0.52 | 0.13 | 0.91 | 0.010* |
| | Post | 3.35 (0.93) | 0–4 | | | | |
| 3- I would consider a well-structured program or training focusing on stem cell science (positive). | Pre | 2.83 (0.91) | 0–4 | 0.65 | 0.38 | 0.92 | <0.001* |
| | Post | 3.48 (0.68) | 2–4 | | | | |
| 4- I think stem cell therapies give rise to ethical controversies (negative). | Pre | 1.29 (1.09) | 0–4 | −0.16 | −0.58 | 0.25 | 0.420 |
| | Post | 1.13 (1.20) | 0–4 | | | | |
| 5- I think stem cell therapies give rise to religious controversies (negative). | Pre | 1.88 (1.10) | 0–4 | −0.75 | −1.23 | −0.27 | 0.003 |
| | Post | 1.13 (1.30) | 0–4 | | | | |
| 6- Government should spend money to support stem cell research (positive). | Pre | 3.38 (0.82) | 1–4 | 0.31 | 0.01 | 0.62 | 0.046* |
| | Post | 3.69 (0.72) | 0–4 | | | | |
| 7- Transitional process of taking stem cell therapy from the laboratory through clinical trials should be encouraged (positive). | Pre | 2.83 (0.93) | 1–4 | 0.44 | 0.12 | 0.76 | 0.009* |
| | Post | 3.27 (0.84) | 1–4 | | | | |
| 8- People should consider the donation of bone marrow for a public bank (positive). | Pre | 2.81 (0.94) | 1–4 | 0.29 | −0.17 | 0.76 | 0.212 |
| | Post | 3.10 (1.23) | 0–4 | | | | |
| 9- People should consider the donation of their babies' umbilical cord blood for a public bank (positive). | Pre | 2.85 (0.10) | 0–4 | 0.42 | 0.00 | 0.83 | 0.049* |
| | Post | 3.27 (1.13) | 0–4 | | | | |
| 10- I am willing to pay money for preserving the umbilical cord blood of my baby in a private bank for later use if a therapeutic need arises (positive). | Pre | 2.63 (1.20) | 0–4 | −0.28 | −0.82 | 0.27 | 0.322 |
| | Post | 2.35 (1.52) | 0–4 | | | | |
| **The total attitude score** | **Pre** | **2.66 (0.56)** | | **0.19** | **0.02** | **0.38** | **0.048*** |
| | **Post** | **2.85 (0.53)** | | | | | |

**Notes.**

Abbreviations: *M*, Mean; *SD*, Standard Deviation; *Min*, Minimum score; *Max*, Maximum score; *CI*, Confidence Interval; *MD*, Mean Difference; *Pre*, Pre-educational intervention; *Post*, Post-educational intervention

\* Significant at $p < 0.05$ based on paired-samples *t*-test.

The total attitude score is the sum of the scores of the ten statements for each participant divided by ten.

(1.13 (1.20) post-intervention *vs.* 1.29 (1.09) pre-intervention, $p = 0.420$) and preserving umbilical cord blood in a private bank (2.35 (1.52) post-intervention *vs.* 2.63 (1.20) pre-intervention, $p = 0.322$) but they were not statistically significant.

**Table 5 Gender differences in mean knowledge and attitude scores pre-and post- educational intervention.**

| Score | Pre-intervention differences between males and females ($n = 48$) | | | Post-intervention differences between males and females ($n = 48$) | | |
|---|---|---|---|---|---|---|
| | Males ($n = 32$) M (SD) | Females ($n = 16$) M (SD) | p-value | Males ($n = 32$) M (SD) | Females ($n = 16$) M (SD) | p-value |
| Total score of stem cell basic knowledge | 2.15 (0.32) | 2.12 (0.24) | 0.734 | 3.15 (0.45) | 2.99 (0.49) | 0.279 |
| Total score of stem cell potential applications | 2.85 (0.66) | 2.28 (0.85) | 0.014* | 3.52 (0.58) | 3.35 (0.61) | 0.369 |
| Total score of stem cell therapeutic uses | 1.82 (0.65) | 1.89 (0.59) | 0.719 | 2.50 (0.86) | 2.35 (0.68) | 0.571 |
| Total score of stem cell research | 1.95 (0.81) | 1.39 (0.95) | 0.036* | 3.30 (0.62) | 3.20 (0.41) | 0.588 |
| **Total knowledge score** | **2.16 (0.27)** | **1.95 (0.30)** | **0.017*** | **3.14 (0.42)** | **3.00 (0.39)** | **0.267** |
| **Total attitude score** | **2.66 (0.60)** | **2.66 (0.50)** | **1.000** | **2.81 (0.52)** | **2.92 (0.55)** | **0.517** |

Notes.
Abbreviations: *M*, Mean; *SD*, Standard Deviation.
* Significant at $p < 0.05$ based on independent-samples *t*-test.

## Gender differences

As shown in Table 5, male students at baseline scored higher knowledge levels in comparison with female students with regard to SC potential applications (2.85 (0.66) *vs.* 2.28 (0.85) respectively, $p = 0.014$) and SC research (1.95 (0.81) *vs.* 1.39 (0.95) respectively, $p = 0.036$). Accordingly, the total knowledge score of males was higher than females (2.16 (0.27) *vs.* 1.95 (0.30) respectively, $p = 0.017$). However, after the intervention, gender differences were not statistically significant.

## DISCUSSION

The current study sheds light on the effectiveness of an interactive educational intervention in improving the knowledge and attitudes of medical students toward SCs, their therapeutic uses, and their potential research applications. The intervention course was conducted for six weeks, and different interactive teaching methods were used. The study results indicated that participants' knowledge about SCs was insufficient in the pre-intervention phase as the mean scores for most knowledge domains and total knowledge were $\leq 2$. These findings are concordant with previous studies that revealed poor knowledge regarding various aspects of stem cells banking, donation, and transplantation among the public, university students, and healthcare providers (*Azzazy & Mohamed, 2016*; *Kaya et al., 2015*; *Lye et al., 2015*; *Perlow, 2006*; *Suen et al., 2011*; *Tuteja, Agarwal & Phadke, 2016*). SC knowledge and attitude scores significantly improved following the intervention course. Post-intervention, participants were more interested in expanding their knowledge about SCs and considered well-structured programs or training courses as a successful approach to improve their understanding of SCs. The participants reported positive attitudes regarding the integration of SC education in medical college curricula after the intervention. This study provides a shred of landmark evidence from the Middle East and North Africa to implement the interactive learning approach in the SC teaching field. The excellent knowledge and attitude regarding SCs could be of great benefits not only to medical students but also to the overall health system as it will reflect on future healthcare providers being more informed and better guided to serve their patients with up-to-date information and improve the decision

making power regarding SCs as an innovative method of therapy (*Perlow, 2006*; *Tork et al., 2017*; *Tuteja, Agarwal & Phadke, 2016*). This study could enhance medical curriculum development and teaching approaches and bridge the gap between basic sciences and clinical practice.

As future health care leaders, medical students represent a source of information, or misinformation, which may influence patients' behaviors and serve as a valuable source of information (*Davies et al., 2002*). This makes the medical school an ideal place to address information misconceptions and emphasize positive attitudes toward SC applications. Therefore, improvements in the region's medical curricula should seriously consider interactive session models and introduce broader and more scientific resources for students in the healthcare field. This is especially true to follow-up on rapidly advancing scientific topics in the medical fields, where relying merely on available evidence from textbooks may introduce delays in transferring knowledge to medical students.

Before conducting the intervention course, the enrolled students reported poor knowledge about SCs, with the lowest scores observed for SC research knowledge, therapeutic uses, and basic knowledge. However, their knowledge regarding the SC potential applications was relatively good. Following the intervention course, the students' knowledge was significantly enhanced with better-reported scores, such as SC research knowledge. Significant improvements also spread to the other addressed knowledge domains regarding SCs, including SC basic knowledge, and that knowledge related to SC potential applications, and to a lesser extent, SC therapeutic uses' knowledge. Previous educational interventions successfully increased the knowledge about SC transplantation and banking among medical, nursing, and law students and showed more positive attitudes toward SC donation following a particular intervention (*Azzazy & Mohamed, 2016*; *Kaya et al., 2015*). Innovative SC education using practical experiments to master SC culture and differentiation techniques were also reported to deepen medical students' understanding of regenerative and translational medicine (*Jin et al., 2018*). After reviewing the educational interventions that were used in other studies to enhance the students' knowledge regarding SCs, we found that their educational interventions were for a shorter duration than ours, and they did not use such interactive teaching methods (*Azzazy & Mohamed, 2016*; *Jin et al., 2018*; *Kaya et al., 2015*). Thus, in our study, we were keen to provide a more comprehensive and detailed interactive teaching course for a longer duration that will cover more topics related to SC education, research, and potential applications, unproven SC therapies and tourism, and bioethics in order to design the interactive teaching course in an innovative way that will be more engaging to the medical students (*Azzazy & Mohamed, 2016*; *Jin et al., 2018*; *Kaya et al., 2015*). Besides, study material developed by our research team could be adopted by other schools interested in establishing similar courses, and our interactive teaching courses could be integrated within curricula.

In the current study, the most common sources of knowledge regarding SCs were lectures followed by social media. Social media has created an opportunity to disseminate information regarding unproven SC-based therapies directly to consumers to legitimize providers and their products by using solid emotional appeals such as patient testimonials (*Lyons, Salgaonkar & Flaherty, 2021*). Other than social media, mass media,

including newspapers, television, and radio, are considered a primary source of scientific communication to the public as it can significantly influence public attitudes toward controversial emerging technologies in regenerative medicine, such as the use of leftover blastocysts as a source for embryonic SCs and genome editing (*Sharpe, Di Pietro & Illes, 2016*). Also, the portrayal of translational SC research in newspapers is highly optimistic and may foster unrealistic expectations regarding clinical translation speed (*Kamenova & Caulfield, 2015*). Medical students should consider other sources for knowledge based on scientific evidence, such as medical journals and conferences. Unfortunately, none of the medical students in our study chose panel discussions as a source for SC knowledge, despite being considered a valuable way to trigger an exchange of viewpoints regarding ethical controversies surrounding SCs (*Arráez-Aybara et al., 2018*). Therefore, medical schools are invited to further invest in students' knowledge about SCs by enhancing exposure to updated medical literature and medical conferences.

This study indicates that the interactive teaching approach effectively improved the levels of knowledge and positive attitudes toward SCs. Furthermore, our interactive learning approach was sufficient to reduce gender gaps in SC knowledge scores, especially the scores related to SC potential applications and SC research since females were significantly less knowledgable than males in the pre-intervention course; however, the gender differences were reduced after the intervention course and became statistically insignificant. Despite the evolving amount of literature indicating the merits of the interactive learning approach, there is still a large gap between educational research and what happens in practice. The traditional didactic lectures still predominate in university classrooms (*Liebert et al., 2016*; *Merideno, Antón & Prada, 2015*; *Saroyan & Snell, 1997*).

Previous studies that compared active learning to the traditional approach using passive learning indicated that interactive teaching methods generally result in longer retention of material, superior problem-solving and higher-thinking skills, more positive attitudes, and higher motivation the students to learn (*DIJK & JOCHEMS, 2002*; *Dodiya, Vadasmiya & Diwan, 2019*; *McKeachie, 1990*; *Wheijen, Jones & Rainer, 2002*). Also, the students found such classes more fun and less tedious, and they were more satisfied with this teaching approach (*DIJK & JOCHEMS, 2002*). Miller et al. reported a statistically significant higher average of students' performance on exams using engaging lectures compared with traditional didactic lectures (*Miller, McNear & Metz, 2013*). Also, the authors observed increased effectiveness of lectures, decreased distractions, and increased students' confidence with the material using interactive teaching methods (*Miller, McNear & Metz, 2013*). Based on our observations while conducting this study and the enrolled students' feedbacks, the experience of interactive teaching technique was interesting for both students and researchers, and many of the students were enthusiastic about more courses designed with this approach. However, many challenges facing the incorporation of interactive teaching methods, including the limited amount of scientific content that could be covered within the class time, the significant efforts of preparation required by the instructor to create the interactive activities, and the effects of this approach on time available for traditional lectures. Given the rapidly growing amount of knowledge and the unlimited access of students to the information through the internet and other technologies,

it may be more important to teach students how to use information rather than learning specific facts. This objective could be achieved by implementation of interactive teaching methods. Thus, medical educators may need to shift the importance of concept over content.

In our study, the participants showed relatively positive attitudes toward SCs before the intervention, and furthermore, their attitudes improved following the intervention even to a lower extent than that observed in knowledge domains. However, negative attitudes related to SC religious controversies actually worsened the post-intervention course. Complex social, legal, ethical, and religious issues arise when emerging biotechnology involves human subjects (*Al-Aqeel, 2005*), especially in conservative societies. However, Islamic teachings have paid attention to disease prevention and health promotion, and it is crucial to focus more on increasing our understanding of how SC applications could advance the health of human beings to facilitate the adoption of these technologies (*Bouzenita, 2017*). Within this context, future SC-related interventions should focus on incorporating religious leaders from the medical community to present their points of view related to scientific facts from ethical, moral, and religious perspectives (*Aksoy, 2005*; *Al-Tabba, Dajani & Al-Hussaini, 2020*; *Fadel, 2012*). The current Statute regarding SC use in developing countries is still unclear and not updated; thus, experts of different political, religious, scientific, and medical aspects who are familiar with laws are invited to develop a more comprehensive juridical system (*Al-Tabba, Dajani & Al-Hussaini, 2020*; *Pourebrahim, Goldouzian & Ramezani, 2020*). However, although the negative attitudes toward ethical controversies surrounding SC therapies worsened following the intervention; the change in mean attitude scores was not statistically significant. Ethical concerns may be tightly connected to religious concerns and can only be mitigated by openly discussing the lack of religious restrictions related to medical improvements.

Notably, our findings regarding religious and ethical controversies call for incorporating bioethics into the medical curriculum when addressing SC-related topics as ethical concerns were reported to be the obstacle that have obscured the proper potential use of SCs for revolutionizing medicine and treatment options in the future (*Hug, 2005*). Medical curricula need to be restructured to include SCs or other emerging technologies in biomedicine and include research and healthcare ethics (*Abdulrazeq et al., 2019*; *Brass, 2009*; *Sarkadi & Schatten, 2012*). Adopting new technologies for patient care is challenging since many ethical dilemmas surround it, and future physicians should be prepared to deal with such dilemmas when they arise (*Curley & Sharples, 2006*).

A few limitations should be mentioned. The sample size was relatively small, and the participants were selected from a single medical school, limiting our results' generalizability. While response and enrolment rates were not optimal, they are considered sufficient among medical students. Data collection in this study was also limited to quantitative methods; utilizing a qualitative approach supported by quantitative methods would be recommended to provide a richer analysis of the phenomena. A parallel-group with no intervention was not utilized which may introduce testing effects and exacerbate the results. The same survey was utilized pre-and post-intervention that was self-reported by the participants, which might explain that the improvements in knowledge and attitudes scores could be by chance

and not due to intervention effects. Finally, this study used an interactive educational approach over a six-week course to improve the students' knowledge and attitudes toward the SC field without comparison with other learning approaches. It would be interesting for future studies to compare the effectiveness of this interactive educational approach with other types of learning, such as flipped classrooms and traditional dedicated lectures.

## CONCLUSIONS

This study demonstrates the crucial role of the interactive teaching approach on medical students' knowledge and attitudes toward SCs, therapeutic uses, and potential research applications. The study results have proven poor knowledge about SCs in the pre-intervention phase, with a significant improvement after the interactive intervention course. As SC utilization becomes more common across specialties, having pre-clinical and clinical curricula educate future physicians on SCs is necessary. Thus, the SC concept such as research skills and SC therapeutic uses should be incorporated into the medical school curriculum to overcome the current shortcomings in SC knowledge and update future physicians with evidence-based medical practice. The study findings also indicate the effectiveness of the intervention course in achieving more positive attitudes toward SCs by the medical students. In addition, the differences in knowledge regarding SCs between males and females have reduced after the intervention course. Thus, the study concluded that an interactive teaching approach might be feasible under the same teaching resources and students' situations, and thus, it could be considered beneficial to the medical students regardless of gender. Educators can capitalize on the available opportunities to improve the areas of the current SC curricula. Although the interactive intervention course was as short as six weeks, the study outcomes were promising. The appropriate intervention methods should be further tested by their implementation on a large scale in most medical subjects. Also, medical educators and schools are calling for integrating new interactive teaching approaches to address the life sciences instead of traditional teaching methods. Further studies with a larger sample are recommended to evaluate the needed curriculum content development, practical teaching approaches, and the most effective practice matters. Moreover, developing educational programs considering social, ethical, legal, religious, and cultural issues are recommended.

## ACKNOWLEDGEMENTS

We would like to thank the faculty of graduate studies for supporting the conduct of this research.

### Funding
The authors received no funding for this work.

### Competing Interests
The authors declare there are no competing interests.

## Author Contributions

- Fayez Abdulrazeq and Khalid A. Kheirallah conceived and designed the experiments, performed the experiments, prepared figures and/or tables, authored or reviewed drafts of the paper, and approved the final draft.
- Abdel-Hameed Al-Mistarehi conceived and designed the experiments, performed the experiments, analyzed the data, prepared figures and/or tables, authored or reviewed drafts of the paper, and approved the final draft.
- Samir Al Bashir conceived and designed the experiments, authored or reviewed drafts of the paper, and approved the final draft.
- Mohammad A. ALQudah and Abdallah Alzoubi analyzed the data, authored or reviewed drafts of the paper, and approved the final draft.
- Jomana Alsulaiman and Mazhar S. Al Zoubi performed the experiments, prepared figures and/or tables, and approved the final draft.
- Abdulwahab Al-Maamari performed the experiments, prepared figures and/or tables, authored or reviewed drafts of the paper, and approved the final draft.

## Human Ethics

The following information was supplied relating to ethical approvals (i.e., approving body and any reference numbers):

All procedures performed in this study involving human participants were reviewed and ethically approved by the Institutional Review Board (IRB) of the research and ethics committee at the University of Science and Technology Yemen-Jordan branch (USTY-Jo) (IRB number, 9/120/2019).

## Data Availability

The datasets generated and analyzed during the current study are available in the Supplementary File.

## Supplemental Information

Supplemental information for this article can be found online at http://dx.doi.org/10.7717/peerj.12824#supplemental-information.

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
