# Peer review of "Effectiveness of interactive teaching intervention on medical students’ knowledge and attitudes toward stem cells, their therapeutic uses, and potential research applications"

_PeerJ, doi:10.7717/peerj.12824_

## Round 0.1 · original submission · Major Revisions

As you will see, both reviewers are enthusiastic about your work and regard it as making an interesting contribution. I have flagged this as a 'major revision', but as you will see the changes are mainly ones of editorial and style and should not present you with major hurdles to overcome.

With one exception. I am slightly concerned about the point raised by reviewer-1 about anonymity and the subsequent use of names to perform the paired tests. Before we can publish this, I need to be reassured that ethical methodology was followed, precisely as written and approved.

The consent form states that the survey is anonymous and confidential, yet participant names are collected in the survey (albeit for a reason). I am unsure about whether this adheres to the ethical approval given for this work.

This point requires careful clarification.

Please indicate in any rebuttal letter your full response to all points raised.

·

Basic reporting

The authors have carried out in interesting study, where they have attempted to evaluate a teaching module on Stem Cells, with pre- and post- course surveys. Together with the surveys, the course design and the ethical approval, I expect this would have been a significant amount of work. The approach of making the effort to collate and publish such data should be commended.

Generally the standard of writing is good. There are some parts than need clarification (see below).

The meanings of the following sections are unclear and should be redrafted:
Line 192 'dully' should be 'duly'
Line 310 - what does 'as a spread' mean?
Line 329 - 'carry and excellent deal'

The following comments could be considered minor:
Abstract background: Line 61 change to...with the potential to alleviate many non-treatable...This avoids the double us of 'treat'.
Line 62: Drop 'The' in 'The medical students...'.
Line 65: Add 'an' to ...effectiveness of (an) interactive
Line 115: Maybe replace 'groundbreaking' with 'novel' or 'pioneering'?

The paper is also formated with the correct sections.

In the other section below I have commented about providing additional references.

Raw data: The additional data is provided as an SAV file. It does not appear that this is one of the recommended journal file formats. SAV appears to be a proprietary SSPS file, but it cannot be opened by other data analysis programs (without add ons). The editorial team should comment on whether this file format is acceptable, however this reviewer cannot examine the raw data in the format provided.

Experimental design

Ethics:
The supplementary documents provided have confirmed alignment to the Declaration of Helsinki.
Participants were all consented and a copy of the consent form in included in the supplementary documents
The consent forms and the paper state that the surveys were anonymous. However, the survey included the student names (line 151- 153). The names were needed to allow the paired t-test, however, this implies that the data collection appears to be inconsistent with the participant consent form. This issue and should be resolved before publication.

Data analysis:

The survey question appears to contain true and false statements. However, this should be mentioned specifically in text: this reviewer has had to infer that from the true/false statements in table 3 (also note that the true/false statements are not explained in the table). There appears to be 10 true statements and 6 false statements (the authors do not provide a numerical breakdown).

In table 3 and the discussion text, it appears that students improve in their agreement with true questions (by more firmly agreeing with the statement and so the score increases) after the training. However, the students also agree stronger with the false questions (by more firmly agreeing with the false statement and so the score increases when it should decrease). This appears to happen in all the false questions (although the reviewer has had to infer this because it is not discussed in the paper). This issue should be addressed before publication.

Line 234-235: two scores 1.84 (SD = 0.63) to 2.45 (SD 0,80) have a p value < 0.001? It's a paired t-test but this does seem like a very low probability. Can the authors confirm this result?.

Validity of the findings

Results and discussion comments:

Line 207 - 'media' be 'social media', as the survey form stated the latter
Line 208 - Data is not presented, but the authors should confirm that the information is in the dataset (see the previous raw data comment).
Line 235-238: Could this sentence be clarified: it’s not clear if the values are referring to the understanding of side-effects, or the understanding of tumor formation potential?


Line 277: The text states that the participant's knowledge is not sufficient in the pre-intervention phases, however by what criteria or standard was this statement made. What defines 'insufficient knowledge'?
Line 283: the authors have attempted to show that an interactive educational approach in SC has benefits to medical students. However, (and with the caveats stated in other paragraphs), the paper doesn't show that the educational approach had to be interactive, or that because it was interactive it was better than a didactic class. The paper should clarify this issue.
Line 285-6: some justification (perhaps a reference) of the statement that an education SC module would improve the overall health service is needed.

Line 303-304: Can the authors state at the end of this sentence what other studies they are referring to, and provide and analysis to demonstrate that this SC module was more 'comprehensive, detailed, and engaging' than those other studies. This question does not appear to be part of the survey.


Line 312- As above, the term media is used, whereas the survey question was related to social media, this should be clarified. The discussion then moves onto 'mass media', which maybe different than social media. Kamenova and Caulfield looked at newspapers (according to their abstract), and so the text should clarify this.
Line 317 - 321: Some evidense (perhaps a reference) is needed for the claim that panel discussions are a 'valuable way to trigger an exchange of viewpoints'.
Line 324 - what are all the knowledge domains? Is it the Basic Knowledge, Potential Applications, Therapeutic Uses and Research? The term 'Knowledge domains' in the educational context could refer to the Cognitive, Affective and Psycho-motor domains of Bloom's hierarchy of learning. Can this be clarified in the text.

Line 334 - Pourebrahim et al. 2020 seemed to conclude that in Iran, SC research and applications should be restricted. This appears to contradict the authors statement in line 338.
Line 366 - Looking at table 5 the gender differences have not vanished, they are still there, but are not statistically significant (perhaps using 'statistically insignificant' may be a suitable term?)
Line 367-369: no evidence is presented that the interactive method of teaching was interesting or that the students were more enthusiastic about a course designed with this approach. There is no data to support the experience of researchers.

Additional comments

As with all reviews, I have tried to take a detailed look at the dataset, analysis and conclusions. I recognised that in some cases my points may be brought on by uncertainity in the text, or my understanding, or a combination of both. I do hope the above issues can be addressed and that with revision this paper could be published.

Reviewer 2 ·

Basic reporting

The authors elaborated the need for their study, the methodology of their course and questionnaire, and the pre vs. post-intervention student outcomes clearly. There are minor errors in English language. For example, the terms "stem cells (SCs)" when combined with a noun such as "biology" or "education" should be replaced with the more appropriate singular of "stem cell." Overall, however, there are no major grammatical errors. The provided tables are well-designed, clearly reflect survey results, and are easy to read.

Experimental design

The authors clearly defined their goal of gauging both attitude and knowledge about stem cell science in medical students before and after a targeted, six-session course. They provided adequate information about ethics, enrollment in the study, demographics, and how metrics were assessed, with the survey being included for our review. The curriculum is elaborated both in prose and in Table 1.

Validity of the findings

The metrics of mean score pre vs. post-intervention seemed appropriate, with the majority of these scores being evaluated with a paired T-test to determine significance. One area where the data analysis may be reconsidered is in table 3, questions 5-13. These true-false questions could be reported in a binary, true/false percentage rather than a sliding scale, which would be a more meaningful metric to assess competency. The authors should also comment more on scores that were relatively low or high in magnitude, even if they improved after intervention. Another discussion that would be relevant to the scope of the article is specialty-related competencies--which fields of medicine were encompassed by the curriculum?

Additional comments

Overall, this appears to be a well-designed curriculum and study. As SC usage becomes more common across specialties, having clinical curriculum that educates future physicians on SCs is increasingly important. To be of interest to a large readership, the most important findings from this study might be a discussion of any shortcomings or gaps (rather than just successes), so that educators at other institutions can capitalize on opportunities to improve areas of their own stem cell curricula.

---

## Round 0.2 · accepted · Accept

Thank you for the careful attention to the issues raised. I am more than satisfied with your response to the ethics question posed and all comments from the reviewers have been effectively addressed. As an advocate of research-led teaching, I found this of interest and congratulate you on this study.